# Core–Shell Polyvinyl Alcohol (PVA) Base Electrospinning Microfibers for Drug Delivery

**DOI:** 10.3390/polym15061554

**Published:** 2023-03-21

**Authors:** Sofía Mares-Bou, María-Antonia Serrano, José Antonio Gómez-Tejedor

**Affiliations:** 1Centre for Biomaterials and Tissue Engineering, CBIT, Universitat Politècnica de València, 46022 Valencia, Spain; 2CIBER-BBN, Biomedical Research Networking Center in Bioengineering, Biomaterials and Nanomedicine, Instituto de Salud Carlos III, 46022 Valencia, Spain

**Keywords:** electrospinning, coaxial electrospinning, Polyvinyl alcohol, controlled drug release

## Abstract

In this study, electrospun membranes were developed for controlled drug release applications. Both uniaxial Polyvinyl alcohol (PVA) and coaxial fibers with a PVA core and a poly (L-lactic acid) (PLLA) and polycaprolactone (PCL) coating were produced with different coating structures. The best conditions for the manufacture of the fibers were also studied and their morphology was analyzed as a function of the electrospinning parameters. Special attention was paid to the fiber surface morphology of the coaxial fibers, obtaining both porous and non-porous coatings. Bovine serum albumin (BSA) was used as the model protein for the drug release studies and, as expected, the uncoated fibers were determined to have the fastest release kinetics. Different release rates were obtained for the coated fibers, which makes this drug release system suitable for different applications according to the release time required.

## 1. Introduction

Electrospinning (ES) is a simple technique for producing nonwoven nanofiber mats or membranes and is a practical and adaptable technique that can produce continuous polymer fibers with diameters ranging from nanometers to microns. Furthermore, this method is versatile and enables the encapsulation of multiple drugs, allowing for the regulation of their release profiles. Additional benefits include this process not necessitating elevated temperatures and the final product being free of any remaining solvent residues [1,2,3,4,5,6,7].

More than 200 polymers have been reported in ES [1]. Natural polymers (collagen, chitosan, gelatin, hyaluronic acid, silk fibroin, etc.) and synthetics (polycaprolactone, polylactic acid, polyglycolic acid, co-polymers of the above, etc.) have been electrospun into nanofibers to form potential scaffolds for biomedical applications [8,9]. Electrospun nanofibers possess numerous remarkable properties, including but not limited to small diameters, substantial surface areas, uniformity, distinctive physical and chemical properties, as well as flexibility [1,10,11,12].

A typical ES configuration (uniaxial electrospinning, U-ES) comprises four essential elements: a capillary tube that holds a polymer solution (or melt), a spinneret or nozzle, a collector, and a high-voltage source. By subjecting a polymer solution to high voltage (usually ranging from 10 to 30 kV), a charge is generated on the surface of the droplet. As the electric field strength intensifies, the solution’s hemispherical surface elongates at the capillary’s tip, eventually forming a cone-like structure called a Taylor cone. Once the voltage is increased even more, a charged jet emerges from the Taylor cone and travels towards the collector. As the jet flows, the solvent evaporates, leaving behind randomly arranged dry polymer fibers on the collector [12,13,14].

Many studies have been conducted of the way the electrospinning process parameters influence the fiber morphology until homogeneous fibers are obtained. The influence of the principal parameters of the electrospinning process have been analyzed for different materials and are classified into solution parameters (solution concentration, solvent, and solvent combinations), process parameters (electric potential difference, distance between needle and collector, needle diameter, and solution flows) and the environmental parameters (temperature and relative humidity). It should be mentioned that all the parameters affect the nanofiber morphology and none of them act independently during electrospinning, so that it is essential to optimize the different parameters to design a nanofiber mat with the desired structure and properties [14,15,16,17].

The primary aim of a drug delivery system is to administer a precise quantity of a specific medication over a predetermined duration based on the medical condition under treatment. Research has been conducted for decades and the drug delivery systems on the market are in different forms and have mechanisms for a multitude of diseases. Even though it is possible to prepare a drug-loaded nanofiber membrane using the U-ES method with a polymer solution and model drug, an initial burst release is an unavoidable issue with this type of blend membrane, which is not ideal for sustained drug delivery. To address this issue, post-treatment of the membranes is necessary, such as cross-linking or chemical modifications. However, these types of post-treatment can result in toxicity and a decrease in biocompatibility. Furthermore, incorporating bioactive molecules or drugs into the thin fiber structure remains challenging since it should not have a negative impact on either the membranes properties or the drug activity. It is possible to adjust the drug release rate by modifying the fiber diameter, porosity, and drug-binding mechanism [18,19,20].

Various advancements have been made in the ES method to improve nanofiber production quality, such as the coaxial electrospinning (C-ES) technique, which produces core-sheath nanofibers. In this process, one polymer nanofiber is surrounded by another, resulting in a combination of both materials’ properties, leading to enhanced overall performance [21,22,23,24]. These systems of encapsulating drugs in polymeric fibers are particularly attractive, with better results than other techniques based on absorption or mixtures, which are frequently used for this purpose [25,26,27,28]. This technique allows the encapsulation of multiple biomolecules. The release kinetics of the encapsulated biomolecules can be controlled by adjusting the system parameters (Figure 1) [29]. The use of a system based on the generation of a C-ES using a coaxial needle can preserve the integrity and bioactivity of the drug because there is no contact with the organic solvent of the polymer in which the drug is encapsulated [30,31,32].

In the present study, bovine serum albumin (BSA) was used as the model protein for the drug release studies. BSA is a protein that is widely and commonly utilized in drug release studies. Its properties, such as its purity, water solubility, affordability, and ease of accessibility, make it a favored choice as a model protein for drug release [34,35].

While some natural polymers are like Extracellular Matrix proteins in native tissue, most have inferior mechanical properties, reduced stability, and faster degradation rates compared to synthetic polymers. Moreover, natural polymers derived from animal sources pose a high risk of contamination due to undesired biological substances present in their molecular structure. Therefore, synthetic polymers are desirable replacements for biomedical applications. Hybrid polymer networks are frequently developed, combining the natural material’s bioactive characteristics with synthetic polymers’ physical attributes. This approach offers a significant advantage by allowing for the customization of the biomaterial’s mechanical and biodegradation properties, providing features that have physiological and bio-functional relevance.

The main material used in this study was Polyvinyl alcohol (PVA), a biocompatible and biodegradable synthetic polymer (Figure 2a), which has been used in various medical applications, including as a vehicle for controlled drug delivery [4,22,36,37,38]. Due to combining PVA with drugs, it can maintain a therapeutic dose during treatment at a specific site, thus avoiding the peaks (translated into the “burst” in the release kinetics).

Our chief aim was to design PVA-based electrospinning membranes for the controlled release of drugs. Manufactured PVA fibrous membranes were produced by uniaxial electrospinning and coating the PVA fibers with other synthetic polymers by coaxial electrospinning. The presence or absence of this coating or shell and its characteristics determine the drug release kinetics.

With no shell structure to the PVA-based microfiber, there is no containment barrier and the drug would be released too rapidly. As the intention is to contain the drug and release its contents either gradually or at controlled rates, it is necessary to implement a structure that surrounds the drug and confines it to the core. The coating’s structure and its degree of degradation control the active ingredient release. Poly (lactic acid) (PLA) and poly (ε-caprolactone) (PCL) were used as the coating polymers in the C-ES membranes.

Poly (lactic acid) (Figure 2b) undergoes hydrolytic degradation in the body and becomes lactic acid monomers that can be metabolized and eliminated from the tissues. This polymer has been used as a material for absorbable surgical sutures for many years and has been proved to be safe and biocompatible [39,40,41].

Poly (ε-caprolactone) (Figure 2c) is a hydrophobic polymer soluble in most organic solvents that exhibits excellent mechanical behavior, undergoes slow degradation and is approved by the Food and Drug Administration (FDA) for biomedical uses [3,21,42].

Without a coating, it is expected to obtain an early release, coated with poly (L-lactic acid) (PLLA), an intermediate release, and with polycaprolactone (PCL), a late release due to its semi-crystalline nature.

## 2. Materials and Methods

Polyvinyl alcohol (PVA) of Mw 13 kDa was purchased from Sigma Aldrich (Darmstadt, Germany). Poly (L-lactic acid) (PLLA) of Mw 280 kDa was provided by NatureWorks (Plymouth, MA, USA) and the polycaprolactone (PCL) of 43 kDa was provided by Polyscience (Niles, IL, USA). Chloroform and N, N-Dimetilformamida (DMF) was purchased from Scharlab (Sentmenat, Spain). Bovine serum albumin (BSA, ≥98%) was purchased from Sigma-Aldrich (Darmstadt, Germany), and the bicinchoninic acid assay (Micro BCA Protein Assay Kit) was purchased from ThemoFisher Scientific (Waltham, MA, USA). A fluorescein isothiocyanate BSA-conjugated (fitcBSA) from Invitrogen (Waltham, MA, USA) was also used.

Phosphate buffer saline without calcium and magnesium (PBS 1×) was the buffer solution, which is prepared manually by dissolving four inorganic salts in mQ water and then adjusting the pH to 7.4 with some drops of 0.2 M sodium hydroxide (NaOH). It consisted of 137 mM sodium chloride (NaCl), 2.7 mM potassium chloride (KCl), 1.8 mM potassium dihydrogen phosphate (KH_2_PO_4_) and 10 mM disodium phosphate (Na_2_HPO_4_). After dissolving the salts to the correct molarity, the pH was adjusted to 7.4 with HCl at a low concentration (0.1 M) and finally stored at 4 °C.

A P-96-well plate from VWR (Radnor, PA, USA) was used to measure absorbance in a VICTOR X3 spectrophotometer reader plate (Wallac 1420 multilabel counter) from PerkinElmer (Waltham, MA, USA).

The spinneret for the U-ES process was a 23-gauge (23 G, OD: 0.642 mm, ID: 0.337 mm) needle from the Hamilton Company (Reno, NV, USA). The spinneret used for the C-ES process consisted of a 25 G inner needle (OD:0.51 mm, ID:0.26 mm) and 16 G outer needle (OD:1.61 mm, ID:1.25 mm) from Tong Li Tech Co (Shenzhen, China).

### 2.1. Preparation of Solutions

PVA was dissolved in mQ water at two concentrations of this polymer, 21% *w/v* and 8% *w/v*. Dissolution was performed at 85 °C for 24 h under magnetic stirring.

PLLA and PCL were dissolved in chloroform and DMF. In the PLLA solutions, the polymer was used at three different concentrations (6, 7.5 and 9% *w/v*) and in two combinations of chloroform and DMF: 80% *v/v* chloroform with 20% *v/v* DMF (named 80:20) and pure chloroform (named 100:0). These solutions were easily completed by keeping the mixture at room temperature (RT) overnight under magnetic stirring. For the PCL dissolutions, the polymer was dissolved at 30% *w/v* due to its low molecular weight (43 kDa) in a 70:30 chloroform:DMF solvent base. This dissolution was completed at RT overnight.

The bovine serum albumin (BSA) protein was always mixed with PVA, except in the control samples. Therefore, as in the PVA solutions, this protein was dissolved at two different concentrations according to its target result, but in all cases the mass ratio of the BSA:PVA was maintained at 1:20. To produce simple microfibers from a mixture of PVA together with BSA, first 21% *w/v* of PVA was therefore dissolved and then BSA was added at 1.05% *w/v*. To produce coaxial fibers with a PVA core and BSA, PVA was first dissolved at 8% *w/v* and then BSA at 0.4% *w/v* was added.

All the solutions produced were stored in a refrigerator at 4 °C between the experiments.

### 2.2. Electrospinning

The U-ES method consists of applying an electric field to a drop of a polymer solution through a needle which is extruded into micrometer-sized threads, while for C-ES, two immiscible solutions are simultaneously driven through two separate feeding channels towards a coaxial needle for the manufacture of fibers with a core–shell structure, where the specific drug will be placed in the core of the fiber and a polymeric material in the fiber shell. The outer polymer solution encapsulates the inner liquid with the biomolecule of interest in an aqueous solution.

In both methods, the solvent evaporates from the fibers on its way to the collector, thus producing solid fibers with a core–shell structure (see Figure 3a). Initially, only water was used inside the fiber with different synthetic polymers dissolved in appropriate solvents for the shell.

The last step of this process involves placing the samples with electrospun membranes in a fume hood overnight to ensure the complete evaporation of the solvent.

The lab equipment is composed of a pump (model NE 1000. New Era Pump Systems, Inc., Toledo St, Farmingdale, NY. USA) connected to the syringe with the polymer solution, a voltage source (OL400W-503, HiTek Power, West Sussex, GB) (Figure 3c) and a metal collector covered by aluminum foil to improve the fiber deposition and correctly manage the samples (Figure 3b). It also includes a humidity and temperature measurement system, and a forced air flow to remove the solvent vapor. In all the experiments, relative humidity was maintained between 40 and 60% and temperature between 22 ºC and 24 ºC.

In C-ES, solutions with very different viscosities prevent the correct formation of the Taylor cone and prevent the correct formation of the fibers. An aqueous PVA solution was used in the internal needle until a viscosity comparable to that of the external solution was achieved. This procedure is compatible with the subsequent incorporation of the drugs in the internal solution, since PVA has already been used in numerous drug delivery systems [4,22,36,37].

PVA 21% *w/v* uniaxial microfibers were produced with the 23 G needle. The core–shell microfibers were produced with a coaxial spinneret composed of inner/outer 25 G/16 G needles. The 25 G inner needle selected was equivalent to an inner diameter of 0.26 mm and an outer diameter of 0.51 mm, leaving an inner ejection area of 0.0531 mm^2^. The 16 G outer needle was equivalent to an inner diameter of 1.25 mm and an outer diameter of 1.61 mm, leaving an ejection gap of 1.023 mm^2^ between the two needles [43].

### 2.3. Characterization of the Microfibers

The fiber morphology was studied by a Field Emission Scanning Electron Microscopy (FESEM, Zeiss, Oxford Instruments, Ultra 55). The fiber diameters were measured with ImageJ software (2022), obtaining 32 measurements from each FESEM image.

### 2.4. Protein Release Assay

Release assays were carried out on fibers loaded with albumin serum bovine (BSA) in aqueous medium using phosphate-buffered saline (PBS). The release kinetics were studied in vitro by ultraviolet radiation spectroscopy to measure protein release at specific timepoints. A series of three replicate electrospun mats loaded with BSA were prepared, to which BSA diluted in distilled water with PVA were in the fiber core. Membranes weighing approximately 10 mg were placed in vials containing 2 mL of PBS. At pre-selected time intervals of up to 30 days, 0.5 mL of supernatant were withdrawn and replaced by 0.5 mL of new and fresh PBS and stored in a freezer at −20 °C until analysis [20,44]. Electrospun membranes without BSA (blank samples) were used as controls.

To determine the initial and final BSA content in the polymer microfibers and solutions, the samples were dissolved in chloroform (2/3 Vol.) and PBS (1/3 Vol.) overnight under magnetic stirring to dissolve the shell polymer and the core PVA. The mixture obtained was then centrifuged for 10 min at 13,000 revolutions per minute (rpm). The resulting aqueous supernatant containing the BSA was then extracted and the quantity of BSA could be determined [45,46].

#### MicroBCA Reaction and Absorbance at 570 nm

Radiation spectroscopy was used to quantify the BSA, which involves the supernatant reacting with the BCA reactant (Micro BCA™ Protein Assay Kit) [20]. The BCA protein assay is a colorimetric method that utilizes a detergent-compatible bicinchoninic acid formulation to detect and quantify total protein. The technique employs BCA as a detection reagent for Cu^+1^, which results from protein reducing Cu^+2^ in an alkaline environment. The chelation of two BCA molecules with one cuprous ion (Cu^+1^) forms a purple-colored reaction product, which displays strong linear absorbance at 562 nm as protein concentrations increase. The result is an extremely sensitive colorimetric protein assay [47].

As detailed in the reagent user manual [47], the working reagent (WR) for this BCA protein assay was prepared by combining 25 parts of Micro BCA Reagent MA and 24 parts of Reagent MB with 1 part of Reagent MC (in a 25:24:1 ratio of MA:MB:MC). The protocol steps were, first of all, pipetting 150 μL of each standard or unknown sample replicate into a 96-well microplate, adding 150 μL of the WR to each well and mixing well with an up-and-down pipette, incubating at 37 °C for 2 h, followed by cooling the plate to room temperature (RT) and measuring the absorbance at 570 nm on a spectrophotometer. Previously, a standard calibration curve from various BSA solutions with known concentrations in PBS was obtained, as the absorbance is proportional to the BSA concentration. (from 1 to 70 mg/mL).

### 2.5. Statistical Analysis

STATGRAPHICS Centurion XIX software was used for the statistical studies. First, the Kolmogorov–Smirnov (K-S) normality test was performed to determine whether the data had a normal distribution and thus ensure the principle of homoscedasticity for the subsequent tests. Two-sample t-Student tests with the Welch variance correction factor were performed. All the statistical tests with a 95% confidence interval (*p* < 0.05) were considered significant. All the data were assigned as mean ± standard deviation (SD). The rest of graphics and trendlines were calculated on Excel 2021 (18.0).

## 3. Results and Discussion

First, U-ES PVA and different C-ES microfibers based on PVA in the core were characterized, except for some with water only, wrapped in a coating of PLLA or PCL. The complexity of this type of characterization lies in the large number of variables and different combinations between the polymer solutions, the core ejection rates and coating solutions, the distance from the needle to the collector, better known as the top-to-collector distance or TCD, and the applied voltage. Second, the release profiles of bovine serum albumin (BSA) protein in the different uniaxial and coaxial microfiber systems previously analyzed were shown.

### 3.1. Microfiber Characterization

#### 3.1.1. Microfiber Distribution and Morphology

Microfibers are qualified by their homogeneity and large-scale distribution, evaluated by their shape and surface and quantified by the variation in their diameters. In all cases, the morphology of the fiber is shown according to some of the electrospinning parameters, leaving the rest of the parameters unchanged. Preliminary studies were carried out to determine the most appropriate conditions for the formation of fibers in each of the materials (results not shown).

##### U-ES Microfibers of PVA 21% as a Function of Flow Rate

The objective was to determine the main differences between PVA 21% U-ES microfiber size and distribution while increasing the electrospinning flow rate from 0.5 to 8 mL/h, doubling the value between consecutive samples. TCD and voltage were kept at 15 cm and 15 kV, respectively. In line with the results obtained, the general visualization of the distribution of the PVA 21% microfibers as deposited during the U-ES process can be assessed as very uniform, thinner, and homogeneous microfibers, with little variation in fiber diameters and no roughness on the fiber surface in all the samples (see Appendix A). All mean and standard deviation (SD) values from the five PVA samples are compared in Figure 4.

After the statistical analysis of these results, it can be seen that there are significant differences between the low and high flow rate samples (0.5, 1 and 2 mL/h and 4 and 8 mL/h, respectively) (Figure 4). It can therefore be deduced that there are significant differences in terms of a slight reduction in the diameter of the microfibers as the flow rate increases, which is in disagreement with most of the studies in the literature, since the higher flow rate is generally associated with increased fiber diameter [6,48]. Regarding the standard deviations obtained, the microfibers can be considered quite homogeneous as the SDs obtained are very small in all cases and especially smaller at high flow rates, from 2 to 8 mL/h.

In short, fine homogeneous fibers were obtained in all cases with highly significant differences between fiber size and a higher flow rate. As the results at high speeds had low SDs, at this point it is important to highlight the interesting bet on applying higher flow rates to increase productivity in terms of a higher production of PVA 21% mesh in a shorter time with a very smooth surface and no apparent roughness in the microfibers.

##### U-ES Microfibers of PVA 21% as a Function of Voltage

The aim of the second analysis of U-ES PVA 21% microfibers was to compare the results obtained by applying a higher potential difference by raising the voltage from 15 to 20 kV while maintaining 15 cm between needle and collector in the samples manufactured at 2 and 4 mL/h (see Figure 5, Appendix A). The results again show a uniform, thinner, homogeneous distribution of U-ES PVA 21% microfibers with no roughness on the fiber surface.

Comparing the results (see Figure 5), the microfiber diameter was seen to increase with a higher applied voltage in the samples manufactured at the 4 mL/h flow rate. This larger diameter could be attributed to an increase in the jet length with the applied voltage [6]. While the fiber diameter at 15 kV diminishes with increasing flow rate (see Figure 5), it appears to be the opposite for 20 kV, so that fiber diameter increases with increasing flow rate as found in the literature [6,48] despite not showing any significant differences in the statistical analysis (*p* value 0.082).

Thinner homogeneous microfibers with no surface roughness were obtained by manufacturing at 20 kV. However, the SD at 20 kV was slightly higher than at the lower voltage, resulting in a slightly lesser homogeneity than PVA meshes at the higher voltage. It thus follows that to obtain more homogeneous, uniform, thinner PVA microfibers, it is better to apply 15 kV than 20 kV.

##### C-ES Microfibers of PLLA 6% 80:20 with a Core of mQ Water as a Function of Inner Flow Rate

Water-loaded fibers with a PLLA polymer coating at low concentration (6% *w/v* in chloroform:DMF 80:20, TCD 15 cm, and voltage 15 kV) were analyzed. This step is based on observing any differences when varying the internal flow rate (the core ejection flow rate) versus maintaining the constant external flow rate (shell ejection flow rate) at 4 mL/h. Samples were thus created from core flow rates at 0.5 mL/h, doubling the rate to 8 mL/h.

The general visualization of the distribution of the microfibers at lower core flow rates (from 0.5 to 2 mL/h) were determined to be not very uniform or homogeneous, so that highly variable fiber diameters were obtained (see Appendix A). The microfiber morphology had a few unwanted elements, including bead formation or garland-shaped structures and a rough fiber surface with pores in the FESEM images. However, as the FESEM image only provides the surface view of the fiber, it is not possible to determine the depth of these pores or if they completely penetrate the coating to the inside of the fiber.

The microfibers at a core flow of 4 mL/h, equal to the shell flow, were homogeneously distributed with little variation in fiber diameter and very few/no undesirable elements, in contrast to the previous samples produced with a lower core flow rate (see Appendix A).

Finally, the microfibers at a higher core flow such that internal flow rate was double the external flow rate (inner flow rate 8 mL/h) had the greatest variation in fiber diameter (see Appendix A). This sample could be considered as completely non-homogeneous since the fiber diameters varied by 94.2% on average. One could thus speak of the presence of two fiber diameter distributions: “fine fibers” between 0.50 and 4 μm and “coarse fibers” between 8 and 10 μm. This could be justified by the large difference between the core flow and sheath flow, giving rise to the bending electrospinning instability that is responsible for the formation of ultrafine fibers by secondary jets [49].

In conclusion, neither uniform nor homogeneous PLLA 6% microfibers loaded with water at the core were obtained. A low core flow rate produced beaded fibers and a porous surface; a certain degree of uniformity could be considered only at the same core and shell flow rate (4 mL/h). The last sample manufactured at an 8 mL/h core flow was discarded due to its high variability in fiber sizes (reflected by its high SD) and a highly significant difference with respect to the rest of the samples with lower core flow rates (see Figure 6).

##### C-ES Microfibers of PLLA 6% 80:20 | PVA 8% as a Function of Inner Flow Rate and Voltage

Four PVA core fibers with a PLLA polymer coating are analyzed here. This step is quite similar to the previous one, observing any differences with the variation of the core flow rate, with a constant shell flow rate (4 mL/h), and compared with the previous study, noting the changes by increasing the voltage to one of the shell–core flow rate combinations. The samples were created with core flow rates at 0.5 mL/h and doubling them to 2 mL/h. This latter condition was repeated by applying 20 kV instead of the 15 kV applied to the rest. The TCD was kept constant at 15 cm.

The general visualization of the distribution of the microfibers at lower core flow rates (from 0.5 to 2 mL/h) can be assessed as neither highly uniform nor homogeneous, as in the previous ones loaded with water, so that the fiber diameters varied widely (see Appendix A). Their microfiber morphology again had a few unwanted elements, including bead formation or garland-shaped structures, and a rough fiber surface with pores in the FESEM images, but it was impossible to determine the depth of these pores. In this case, the bead formation increased with the core flow, as mentioned previously [1,48]. All the mean and standard deviation (SD) values of these three PLLA-PVA shells-core samples are compared in Figure 7.

The microfibers at 2 mL/h core flow rate had differences in terms of distribution and homogeneity when manufactured at 15 kV (Appendix A) and 20 kV (Appendix A) with a TCD of 15 cm. The mesh coaxial fiber distribution, homogeneity, size variability and, above all, the formation of large pearls significantly worsened when voltage was raised, so that this last sample made at 20 kV was discarded in further assays.

In the statistical analysis of these results, there were significant differences between the samples at the lowest flow rates (0.5 mL/h) and those at higher flow rates (1 and 2 mL/h) (Figure 7), indicating significant differences in terms of a slight increase in the diameter of the microfibers as the flow rate was raised as reported in previous studies [6,48].

Non-uniform and non-homogeneous microfibers were obtained, as well as highly significant differences between the fiber size with the increased flow rate. At low core flow rates, beaded fibers and a porous surface were formed, while the last sample at 20 kV was discarded due to its extreme bead formation, unacceptable in a microfiber net.

##### PLLA 9% 80:20 | PVA 8% C-ES Microfibers as a Function of Flow Rate and Voltage

The PVA 8% core fibers with a PLLA polymer coating at a higher concentration (9% *w/v*) are analyzed in this section, distinguishing between the solvent combinations used in the polymer solution, i.e., this section describes the samples of PLLA 9% dissolved in chloroform:DMF 80:20, while the samples based on polymer solution only in chloroform, i.e., 100:0 are dealt with in the following section.

In this section, we attempted to detect differences in the microfibers by changing the core flux, the external flux, and, in some cases, the distance and voltage at the same time. Core fluxes from 0.5 to 2 mL/h, coating fluxes at 4, 6 and 8 mL/h and certain combinations at 15/20 cm and 15/20 kV were thus analyzed.

The distribution of the coaxial microfibers deposited during electrospinning with a constant shell flow rate of 4 mL/h and core flux rate variation (Appendix A) and constant core flow at 0.5 mL/h as well as shell flux variation (Appendix A) can be considered generally very uniform and homogeneous, with little variance in the fiber diameters, except for the sample manufactured at 1 mL/h core flux (Appendix A), whose fiber distribution was not considered uniform as it displayed too much variation in fiber size. All the mean and SD values of these samples are compared in Figure 8 and Figure 9.

The microfiber morphology of these PLLA 9% 80:20 and PVA 8% core fibers had rough fiber surfaces with pores in the FESEM images, but without bead formations. The thinnest and smoothest surface was obtained in the sample produced at 2 mL/h core flux at 20 kV (Appendix A) and 6 mL/h shell flux at 20 kV (Appendix A).

In the coaxial microfibers manufactured at 0.5 mL/h (Appendix A) and 1 mL/h (Appendix A) at 20 cm and 20 kV, one could speak of the presence of two or three fiber diameter distributions, respectively, with “fine fibers” of between 0.9 and 2.1 μm and “intermediate fibers size” of around 3 μm in the first case and in the second case, “fine fibers” of between 0.1 and 2 μm, “intermediate fibers size” of between 2 and 3 μm, and “coarse fibers” of around 4 μm.

After the statistical analysis of these results, there were significant differences when the distances and voltages were changed and also when the flow rate was modified. When increasing both the TCD from 15 to 20 cm and the voltage from 15 to 20 kV, larger fiber sizes were obtained, as well as a greater diversity in size (Figure 8a). However, in the case of the 6 mL/h shell flux, the microfiber size remained constant, increasing the voltage from 15 to 20 kV (Figure 9b).

On the other hand, there were significant differences between the samples at a lower core flow rate (0.5 and 1 mL/h) and a higher core flow rate (2 mL/h) (Figure 8b); also between the samples at low shell flow rates (4 mL/h) and those with a higher shell flow rate (6 and 8 mL/h) (Figure 9a). In the first case, there were significant differences in terms of a slight reduction in the diameter of the microfibers as the flow rate was increased, while the opposite occurred in the second case, which is in agreement with many of the published studies [6,48]. There were also no significant differences between the low core flux (0.5–1 mL/h) and high shell flux (6 and 8 mL/h); therefore, a constant fiber size between these fluxes was considered.

In conclusion, a uniform and homogeneous distribution together with a rough porous surface were mainly obtained from PLLA 9% microfibers loaded with PVA 8% at the core from chloroform:DMF 80:20 solvent in the polymer solution. The smoother fiber surface obtained at 0.5–6 mL/h core–shell flow rates, 15 cm and 20 kV deserve special attention.

##### PLLA 9% 100:0 | PVA 8% C-ES Microfibers as a Function of Flow Rate and Voltage

As in the previous study, the following samples were compared in terms of the variation in the core flow rate, applied voltage and shell flow rate. However, these subsections had a different polymeric coating solution, since only chloroform organic solvent was used. A core flow rate of 0.5 mL/h to 2 mL/h, a shell flow rate of 4 mL/h to 8 mL/h, and a voltage of 15 and 20 kV were applied, maintaining TCD at 15 cm in all cases.

In the fibers with core flux rate variation (Appendix A) and shell flux variation (Appendix A), the distribution of the coaxial microfibers deposited during electrospinning can generally be considered very uniform and homogeneous. All the mean and SD values of these samples are compared in Figure 10 and Figure 11.

The microfiber morphology had an apparently rough surface in all cases, and it could even be concluded that the porous voids did not penetrate the entire layer of the porous coating. In addition, ribbon-shaped and garland-shaped bead formations were observed in all cases, especially in one case in which a droplet was probably projected and some kind of porosity was unintentionally generated (see Appendix A).

After the statistical analysis of these results, there were significant differences when the voltage was changed and also when the flow rate was modified. On raising the voltage from 15 to 20 kV, smaller fiber sizes were obtained (Figure 10b and Appendix A), the opposite of the 80:20 solvent sample results.

On the other hand, there were significant differences between the samples when increasing the core flow rate (Figure 10a) and also between the samples produced at the lowest shell flow rate (4 mL/h) with respect to higher shell flow rate (6 and 8 mL/h) (Figure 11). In the first case, there was a curious contradiction, since there were significant differences in terms of a slight reduction in the fiber diameter as the flow rate was raised from 0.5 to 1 mL/h, also a larger fiber diameter as the flow rate rose from 0.5 or 1 to 2 mL/h. In the second case, significant differences were obtained in the larger fiber size by increasing the external flow from 4 to 6 and 8 mL/h [6,48].

The fiber size could be considered constant among the high velocities.

In conclusion, uniform and homogeneous distribution, rough porous surface and bead formation were mainly obtained from chloroform solvent in polymer solution (100:0).

##### PLLA | PVA 8% C-ES Microfibers as a Function of Concentration and Solvent Ratio

The best fabrication procedure obtained for all the above samples was an internal flow rate of 0.5 mL/h versus an external flow rate of 4 mL/h at a distance of 15 cm and a voltage of 15 kV. A comparison of the diameter of the PVA-based core coaxial fibers is shown below by coating with PLLA at 6% 80:20 (Appendix A), 9% 100:0 (Appendix A) and 9% 80:20 (Appendix A). After the statistical analysis of these PLLA-PVA shell–core fibers, there were significant differences in the increase in fiber size on increasing the PLLA concentration and using a mixed solvent system (Figure 12).

Conflicting results have been reported in the literature on the effect of polymer concentration on fiber dimensions, but most of them indicate that a high polymer concentration results in larger diameters, while a low polymer concentration produces smaller fibers [1,2], as happened in this case.

After analyzing all the samples of microfibers with a PLLA coating, it was determined that the 9% *w/v* polymer solution reduced the bead-like structures and although the microfiber surface porosity was maintained, they were much smaller in size. The samples produced at higher flow rates provided a favorable result, indicating a more efficient sample production system.

#### 3.1.2. Microfiber Surface

In the analysis of the microfiber surface, the main difference observed was the formation of nanopores in the fiber sheath. However, it could not be determined whether these holes penetrate the entire shell (Figure 13). The formation of these structures is interesting in regard to the analysis of the release of charged substances in the core.

First, the solvent plays a vital role in the fabrication of highly porous microfibers. This may occur when a polymer is dissolved in two solvents, when one solvent acts as a non-solvent. The different evaporation rates of the solvent and non-solvent lead to phase separation and result in the fabrication of highly porous electrospun microfibers [6]. In this case, chloroform and DMF were used as mixed solvents for the shell polymer solution. The porous surface could thus be attributed to the difference between both volatilities, even though their conductivities and dipole-moments are also important and influential. In this case, the DMF volatility or boiling temperature is lower than that of chloroform, which leads to faster evaporation and thus to the formation of spaces between the polymeric structure, which leads to the formation of these nanopores (see Figure 13a) [2,10,31,50].

Second, this effect can be attenuated as the applied voltage is increased, so that the fibers undergo greater deformation during manufacturing process, and this causes the solvents to mix to a greater extent, resulting in almost non-existent nanopore spaces on the surface of the microfibers (see Figure 13b) [6]. The same justification could be used when the average diameter of porous core–shell nanofibers is slightly reduced with increasing core feed rates [51].

In this study, it was seen that the PLLA-coated samples had the ability to form rough or porous surfaces, although other studies discovered this surface property in PCL coatings [52], which explains why porosity decreases as the polymer concentration is increased.

### 3.2. BSA Release

Table 1 provides the details of the six samples chosen for the release test, in which the first five samples were coated (C-ES) and the last sample was uncoated (U-ES), showing polymer concentration, solvent ratio (chloroform:DMF) and BSA concentration.

The release results are shown as the amount of BSA released in mg (Figure 14) and amount of BSA released in μg per mg of the initial mass used of each sample for the release test (Figure 15). The results are shown in micrograms of BSA released per milligram of sample in order to be able to compare all the samples, whether coated or uncoated. As mentioned in the Section 2, the concentration chosen for the uncoated PVA microfibers with 1.05% *w/v* BSA in the PVA sample was precisely so that it would contain approximately the same amount of BSA per mg of sample as the rest. This could also justify this representation of release per mg of sample, as in Figure 15, by the fact that this factor is what really matters in a practical application: how much is released based on the weight of the sample.

First, the highest release occurred in the uncoated PVA fibers, and also very quickly as they released 80 μg/mg in 3 days, and thereafter only 10 μg/mg. This was to be expected, as the coating encapsulated the BSA inside the fiber in the other samples and slowed down the release.

Second, in the release of the rest of the coaxial microfibers, i.e., those with a polymeric coating, samples PLLA9%80:20, PLLA7.6% and PCL released the least. The PCL sample is clear because it has no pores and has a homogeneous coating, which retains the BSA inside the fiber and means that the release is low and slow. The PCL sample is also expected to have a slower release, because it has a slower degradation time and so hardly degraded in the 30 days of the trial, so the release could be expected to be slow, since it only occurs by diffusion through the polymer matrix. Considering the PCL and PLLA degradation, the PCL has much longer degradation times due to their semi-crystalline nature, while the electrospun PLLA nanofibers were completely non-crystalline but had highly oriented chains and a lower glass transition temperature than the cast film [48]. The release results obtained suggest that it is the latter, i.e., that the pores that appear on the surface of the microfibers do not pass through the coating [51]. However, a more detailed analysis would be necessary to obtain a clearer conclusion. The results of this test disagree with those obtained by Tiwari et al. (2010), where PLLA-coated microfibers showed a relatively low release while the porous PCL showed a rapid burst release [52].

Third, we discovered PLLA9% and PLLA7.5%80:20 samples with intermediate kinetics. Based on the FESEM images, the appearance of the pores is also unknown, but the faster initial release suggests that these pores penetrate the entire fiber coating until they reach the core, and that is why they are released more quickly. However, as in the previous case, a more detailed analysis would be necessary to obtain a clearer conclusion.

Finally, another important factor to consider is the thickness of the fiber. Under same ES conditions, it could be expected that the higher the polymer concentration, the thicker the coating (samples PLLA9%80:20 and PLLA7.5%80:20; and PLLA9% and PLLA7.5%). A thicker coating implies a slower release, which is true between samples PLLA9%80:20 and PLLA7.5%80:20, since PLLA9%80:20, which has a higher polymer concentration, has a slower release than PLLA7.5%80:20. However, the opposite occurs between PLLA9% and PLLA7.5%, with PLLA7.5% releasing slower than PLLA9%. Another interesting aspect to consider in further trials is the comparison of samples at different flow rates, as it has been seen in other studies that higher core feeding rate results in faster protein release [22].

## 4. Conclusions

The conclusions obtained can be classified in terms of the electrospinning technique, the analysis of the microfibers depending on the electrospinning technique and the results of the release of the BSA encapsulated in coaxial microfibers.

Different electrospinning methods have been developed and a wide variety of samples have been produced according to polymeric samples, electrospinning techniques, materials, and conditions. On the one hand, thin, smooth uniform PVA microfibers were obtained by electrospinning with an average fiber size of approximately 0.30 μm. On the other hand, thicker, rougher, porous and not so uniform coaxial microfibers with a PLLA coating and PVA core were obtained by electrospinning with a fiber size of between 0.50 μm and 2.50 μm, depending largely on the concentration of the coating polymer, distance between the tip to the metallic collector and the applied voltage. PCL coated and U-ES PVA core microfibers were also manufactured.

A drug release system was assayed by a protein BSA-based model that obtained a smooth release in all the BSA-loaded microfiber samples, with no burst release. In terms of release kinetics, the PVA microfibers released more drug in a shorter time. In the release from the coaxial fibers, the PLLA coating at higher concentrations from a simple chloroform solvent released first, the PLLA coating at a lower concentration and mixed solvent together with DMF released second, and the PCL coating released third. These results were attributed to the non-polymeric coating of the first sample, the roughness and porosity obtained on the surfaces of the microfibers, and the semi-crystalline nature of PCL, respectively.

## Figures and Tables

**Figure 1 polymers-15-01554-f001:**
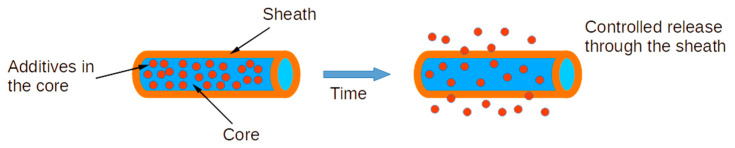
Scheme of the encapsulation and release procedure [33].

**Figure 2 polymers-15-01554-f002:**
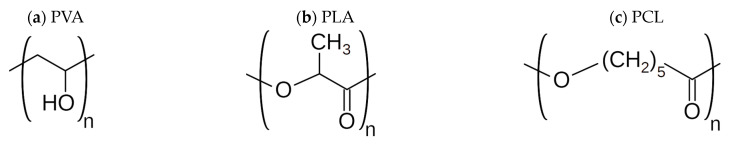
Chemical structure of polymers: (**a**) PVA, (**b**) PLA, and (**c**) PCL.

**Figure 3 polymers-15-01554-f003:**
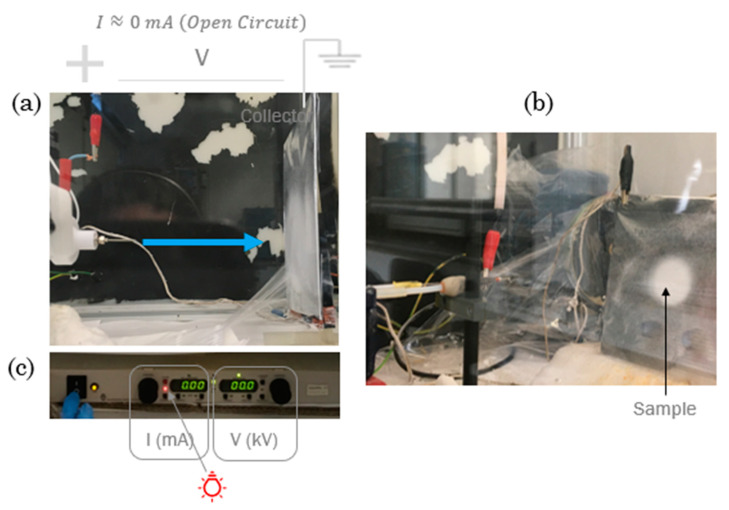
Coaxial electrospinning assembly (**a**), sample visualization (**b**), and high voltage source, where intensity (I) in mA and voltage (V) in kV are displayed (**c**).

**Figure 4 polymers-15-01554-f004:**
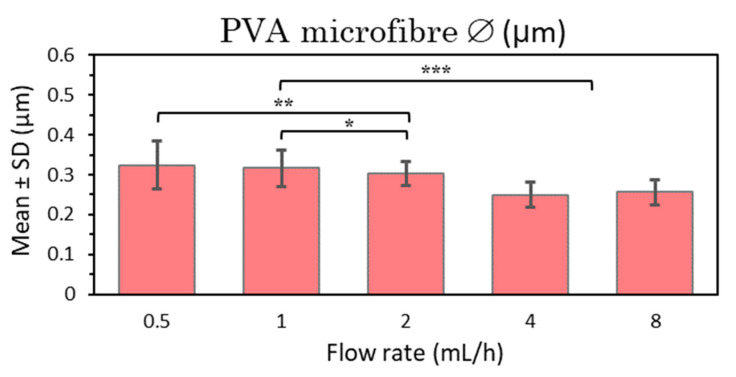
Mean and standard deviation of the diameters of 21% PVA microfibers manufactured at 15 cm, 15 kV, and 0.5, 1, 2, 4, and 8 mL/h., at (*) *p* < 0.05, (**) *p* < 0.01, (***) *p* < 0.001 statistically significant differences.

**Figure 5 polymers-15-01554-f005:**
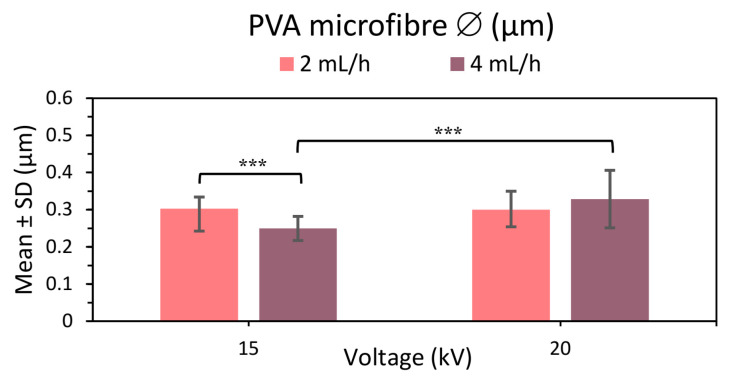
Mean and standard deviation of the diameters of 21% PVA microfibers manufactured at 2 and 4 mL/h and applying 15 and 20 kV, at (***) *p* < 0.001 statistically significant difference.

**Figure 6 polymers-15-01554-f006:**
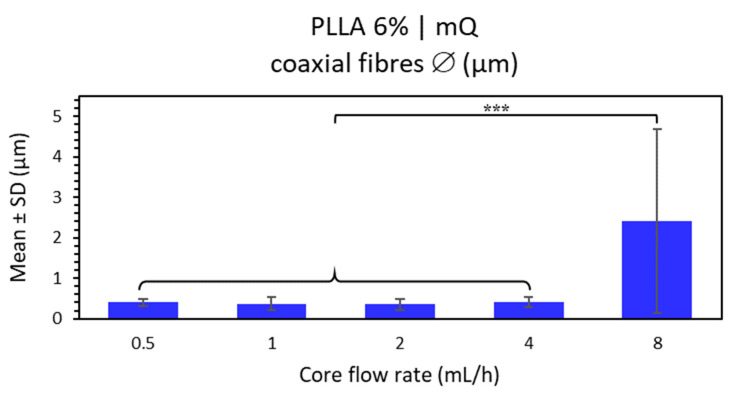
Mean and standard deviation of the diameters of PLLA 6% chloroform:DMF 80:20 (shell) and water mQ (core) coaxial microfibers manufactured at internal flow of 0.5, 1, 2, 4 and 8 mL/h, external flow of 4 mL/h, with 15 cm and 15 kV, at (***) *p* < 0.001 statistically significant difference.

**Figure 7 polymers-15-01554-f007:**
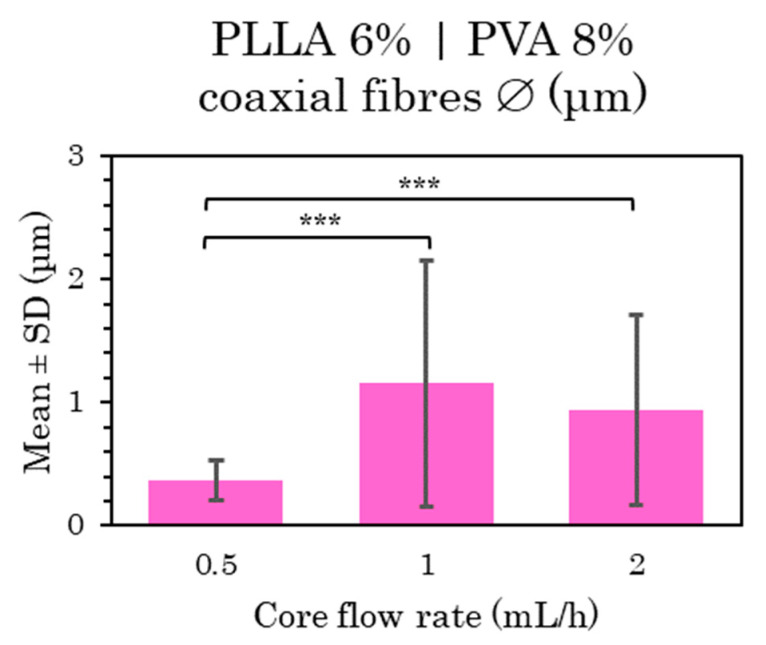
Mean and standard deviation of the diameters of 6% PLLA (shell) and 8% PVA (core) coaxial microfibers manufactured at 0.5, 1 and 2 mL/h, with 15 cm and 15 kV, at (***) *p* < 0.001 statistically significant difference.

**Figure 8 polymers-15-01554-f008:**
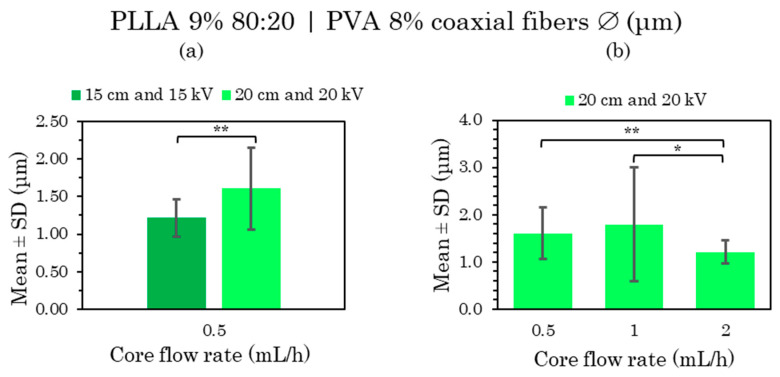
Mean and standard deviation of the diameters of 9% PLLA 80:20 (shell) and 8% PVA (core) coaxial microfibers manufactured at 0.5 mL/h with 20 cm and 20 kV compared with 15 cm and 15 kV (**a**), and at 0.5, 1 and 2 mL/h with 20 cm and 20 kV (**b**). Shell flow rate 4 mL/h in both cases, at (*) *p* < 0.05, (**) *p* < 0.01 statistically significant differences.

**Figure 9 polymers-15-01554-f009:**
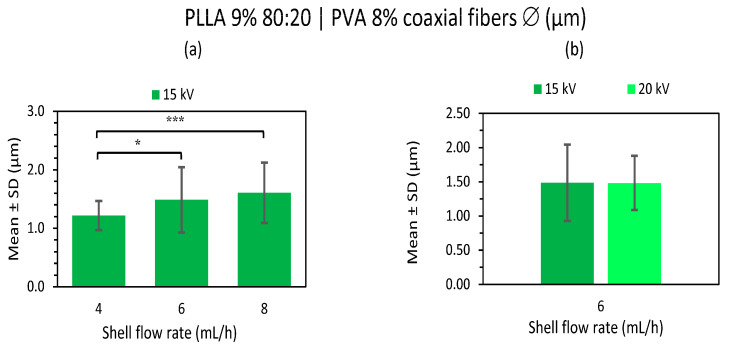
Mean and standard deviation of the diameters of 9% PLLA 80:20 (shell) and 8% PVA (core) 0.5 mL/h coaxial microfibers manufactured at 4, 6 and 8 mL/h shell flow rate with 15 cm and 15 kV (**a**), and at 6 mL/h shell flow rate with 15 kV compared with 20 kV (**b**). Core flow rate 0.5 mL/h in both cases, at (*) *p* < 0.05, (***) *p* < 0.001 statistically significant differences.

**Figure 10 polymers-15-01554-f010:**
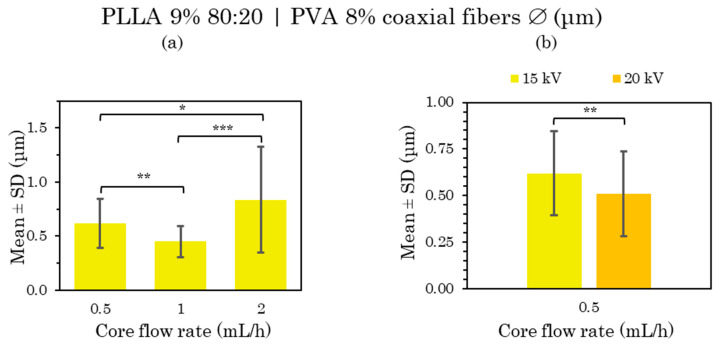
Mean and standard deviation of the diameters of 9% PLLA 100:0 (shell) and 8% PVA (core) coaxial microfibers manufactured at 0.5, 1 and 2 mL/h core flow rate with 15 cm and 15 kV (**a**), and at 0.5 mL/h core flow rate with 15 kV compared with 20 kV (**b**). Shell flow rate 4 mL/h in both cases, at (*) *p* < 0.05, (**) *p* < 0.01, (***) *p* < 0.001 statistically significant differences.

**Figure 11 polymers-15-01554-f011:**
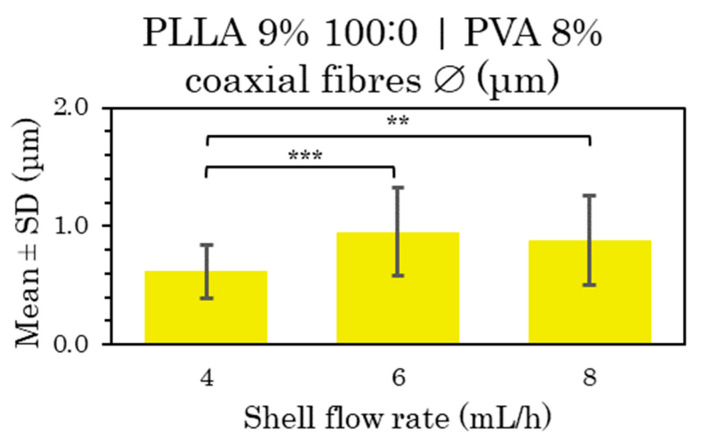
Mean and standard deviation of the diameters of 9% PLLA 100:0 (shell) and 8% PVA (core) coaxial microfibers manufactured at 4, 6 and 8 mL/h shell flow rate with 15 cm and 15 kV. Core flow rate 0.5 mL/h in both cases, at (**) *p* < 0.01, (***) *p* < 0.001 statistically significant differences.

**Figure 12 polymers-15-01554-f012:**
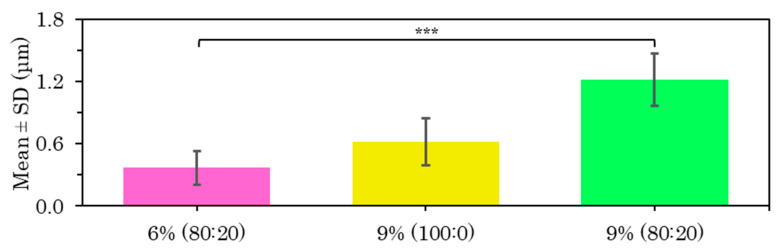
Diameter results for coaxial microfibers of PLLA shell loaded with PVA 8% core manufactured with shell–core flow 4–0.5 mL/h at 15 cm and 15 kV with the mean and SD for 6% and 9% concentration in 80:20 chloroform:DMF solvent and 9% concentration in 100:0 solvent, at (***) *p* < 0.001 statistically significant difference.

**Figure 13 polymers-15-01554-f013:**
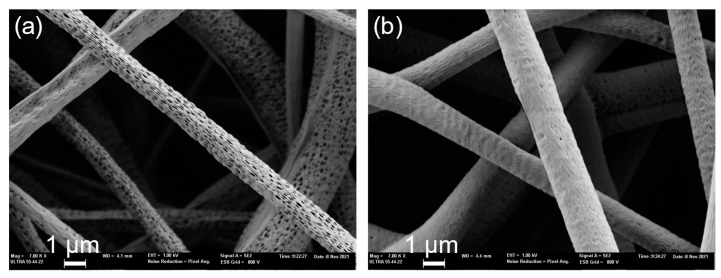
Coaxial microfibers of PLLA 9% chloroform:DMF 80:20 shell and PVA 8% core manufactured with shell–core flow 6–0.5 mL/h at 15 cm and 15 kV (**a**) and 20 kV (**b**) by FESEM.

**Figure 14 polymers-15-01554-f014:**
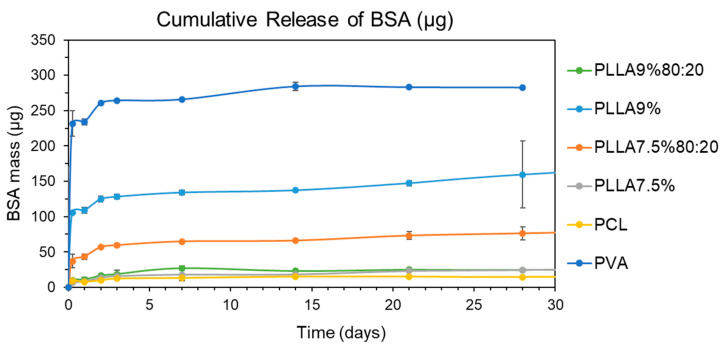
Cumulative release in mass of BSA.

**Figure 15 polymers-15-01554-f015:**
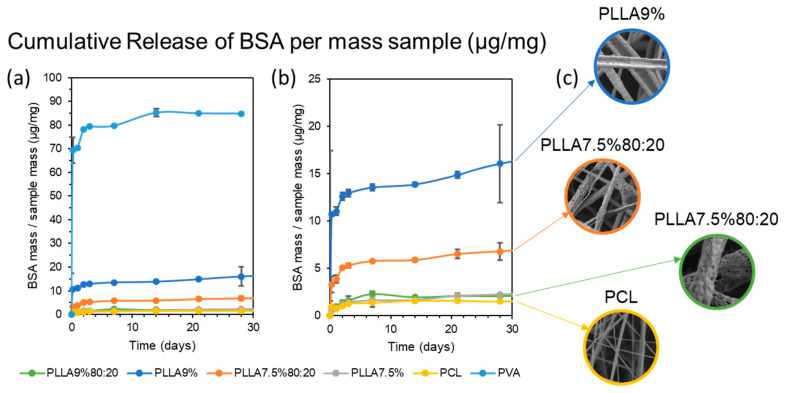
Cumulative release of BSA from the six samples per each sample mass (**a**), zoom for the five coaxial fiber samples (**b**) and visualization of samples by FESEM (**c**).

**Table 1 polymers-15-01554-t001:** Legend of the samples included in the BSA release test with their composition and nature (coaxial or uniaxial).

Sample Notation	Composition of the Microfiber Shell | Core
PLLA9%80:20	PLLA 9% (80:20) | PVA 8% + BSA 0.4%
PLLA9%	PLLA 9% (100:0) | PVA 8% + BSA 0.4%
PLLA7.5%80:20	PLLA 7.5% (80:20) | PVA 8% + BSA 0.4%
PLLA7.5%	PLLA 7.5% (100:0) | PVA 8% + BSA 0.4%
PCL	PCL 30% (70:30) | PVA 8% + BSA 0.4%
PVA	PVA 21% + 1.05%

## Data Availability

The data presented in this study are available in Appendix A and on request from the corresponding author.

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
