# Peer review of "Core–Shell Polyvinyl Alcohol (PVA) Base Electrospinning Microfibers for Drug Delivery"

_polymers, 2023, doi:10.3390/polym15061554_

Round 1

Reviewer 1 Report

In this work, the electrospinning membranes were produced with different coating structures for drug-release application. The best conditions for the manufacture of the fibers were also studied and their morphology was analyzed as a function of the electrospinning parameters. The results indicated different release rates were obtained for the coated fibers, which makes this drug release system suitable for different applications. In fact, there are many studies about electrospinning membranes for drug-release application. This study presented in this manuscript are not novel. There are no detailed contents for discussing the mechanism and its novelty. It makes no sense for this work. In my opinion, I think this paper is not suitable for publication in polymers.

Author Response

The authors would like to thank the reviewers and editors for the time they spent reviewing our manuscript and for their careful assessment and comments. The suggestions were extremely helpful in enhancing the quality of the manuscript. We have tried to answer all your questions/remarks to the best of our knowledge. The responses to the suggestions made are as follows.

All the best

Reviewer comment #1: In this work, the electrospinning membranes were produced with different coating structures for drug-release application. The best conditions for the manufacture of the fibers were also studied and their morphology was analyzed as a function of the electrospinning parameters. The results indicated different release rates were obtained for the coated fibers, which makes this drug release system suitable for different applications. In fact, there are many studies about electrospinning membranes for drug-release application. This study presented in this manuscript are not novel. There are no detailed contents for discussing the mechanism and its novelty. It makes no sense for this work. In my opinion, I think this paper is not suitable for publication in polymers.

Answer #1:

You are right when you say that there are many works in the literature related to the application of electrospun membranes for drug delivery. And although there are some commercial products on the market that use this technology for drug delivery in some applications, the problem is not completely solved, and there are still many challenges to overcome and work to do to understand the manufacturing processes of electrospun membranes and release processes from them.

In this way, this work intends, first of all, to analyze the morphology of the fibers manufactured by coaxial electrospinning, depending on the electrospinning parameters, with the aim of finding the best manufacturing conditions.

Secondly, it aims to show the results of the release of a model protein (BSA) from the developed electrospun membranes.

For all these reasons, we consider that the article has sufficiently novelty to be published in the Polymers journal, and in particular in the special issue of "Electrospun Composite Nanofibers for Functional Applications II".

Reviewer 2 Report

Excellent work in the field of nanofibres for drug delivery. The entire manuscript is very well written but it is highly statistics oriented. All mean deviation figures are very much highlighted and the rest of the images are kept in supplementary material which should not be the case. Instead, it should be reversed.   Kindly, revised the manuscript accordingly. What about the effect of various drugs? So far only one has been tried. Some comments are also marked in the manuscript. 

Author Response

The authors would like to thank the reviewers and editors for the time they spent reviewing our manuscript and for their careful assessment and constructive comments. The suggestions were extremely helpful in enhancing the quality of the manuscript. We have tried to answer all your questions/remarks to the best of our knowledge. We tried to address all the queries. In order to lead the reviewers to the modifications incorporated in the revised manuscript, all changes have been highlighted using Microsoft Word's Track Changes tool. The responses to the suggestions made are as follows.

All the best

Reviewer comment #1: Excellent work in the field of nanofibres for drug delivery. The entire manuscript is very well written but it is highly statistics oriented. All mean deviation figures are very much highlighted and the rest of the images are kept in supplementary material which should not be the case. Instead, it should be reversed. Kindly, revised the manuscript accordingly.

Answer #1:

Thank you very much for your positive comment about the quality and interest of the paper.

As the reviewer comments, we have emphasized the analysis of the mean values of fiber diameter based on the parameters analyzed. We believe that this is a good way to understand and analyze the morphology of the fibers as a function of the different parameters studied. To do otherwise would have greatly increased the length of the paper since the supplementary material contains 29 figures. The information provided by these figures is very interesting and relevant to the article, but we believe it is more convenient to keep these figures in the supplementary material, so as not to lose the main paper arguments.

Reviewer comment #2: What about the effect of various drugs? So far only one has been tried.

Answer #2:

We have tested only the BSA release. This is a protein that is used in many references as a model drug for the analysis of drug release processes. We have introduced the following paragraph in the introduction section to clarify it:

“In the present study, bovine serum albumin (BSA) was used as the model protein for the drug-release studies. BSA, is a protein that is widely and commonly utilized in drug release studies. Its properties, such as its purity, water solubility, affordability, and ease of accessibility, make it a favored choice as a model protein for drug release [34,35].”

In future works, we are planning the analysis of the release of other drugs, applied to ocular diseases, but this was not the objective of the present work.

Reviewer comment #3: Some comments are also marked in the manuscript. 

Answer #3:

Thank you very much again for your comments. We have reviewed these comments, and have made the following changes in the paper:

Page 3, line 101

We have replaced the sentence “As the coating degrades, its content, and therefore the drug, is released at a controlled rate determined by the coating’s degree of degradation” by “The coating’s structure and its degree of degradation control the active ingredient release.”

Page 5, line 175: Is this pump code or connecting cable

We have removed “RS-232”. The pump model is NE 1000. RS-232 stands for the kind of connection cable, but it is not the pump code.

Page 9, line 354

A missing dot is included at the end of the sentence: “The samples were created with core flow rates at 0.5 mL/h and doubling them to 2 mL/h.”

Page 10, line 374

A missing coma is included in the sentence: “In the statistical analysis of these results,”

Page 15, line 524

The word “during” has been introduced instead of “in their”.

Page 15, line 529

A missing coma is included in the sentence: “In this study,”

Round 2

Reviewer 1 Report

This study aims to design and develop a novel electrospun membranes for controlled drug-release applications. After revision, I think this paper is accepted for publication in Polymers.